# Video-Assisted Stylet Intubation with a Plastic Sheet Barrier, a Safe and Simple Technique for Tracheal Intubation of COVID-19 Patients

**DOI:** 10.3390/healthcare10061105

**Published:** 2022-06-14

**Authors:** Ching-Hsuan Huang, I-Min Su, Bo-Jyun Jhuang, Hsiang-Ning Luk, Jason Zhensheng Qu, Alan Shikani

**Affiliations:** 1Department of Anesthesia, Hualien Tzuchi Hospital, Hualien 97002, Taiwan; velociraptor4226@gmail.com (C.-H.H.); 100311034@tzuchi.com.tw (I.-M.S.); 100311050@tzuchi.com.tw (B.-J.J.); 2Bio-Math Laboratory, Department of Financial Engineering, Providence University, Taichung 43301, Taiwan; 3Department of Anesthesia, Critical Care and Pain Medicine, Massachusetts General Hospital, Harvard Medical School, Boston, MA 02114, USA; jqu@mgh.harvard.edu; 4Division of Otolaryngology-Head and Neck Surgery, LifeBridge Sinai Hospital, Baltimore, MD 21215, USA; ashikani@marylandent.com; 5Division of Otolaryngology-Head and Neck Surgery, MedStar Union Memorial Hospital, Baltimore, MD 21218, USA

**Keywords:** difficult airway, video-assisted intubating stylet, Shikani technique, video-laryngoscopy, plastic sheet, barrier enclosure, COVID-19, Omicron, pandemic

## Abstract

As the COVID-19 pandemic evolves, infection with the Omicron variants has become a serious risk to global public health. Anesthesia providers are often called upon for endotracheal intubations for COVID patients. Expedite and safe intubation can save patient’s life, while minimizing the virus exposure to the anesthesia provider and personnel involved during airway intervention is very important to protect healthcare workers and conserve the medical work force. In this paper, we share clinical experience of using a video-assisted intubating stylet technique combined with a simple plastic sheet barrier placed over the patients’ mouth for tracheal intubation during the Omicron crisis in Taiwan. We demonstrated that the use of an intubating stylet combined with plastic sheet barrier is swift, safe, and accurate in securing the airway in patients with COVID-19.

## 1. Introduction

The spread of COVID-19 was temporarily contained by the end of 2021 in Taiwan after many lives were lost since the pandemic started in December 2019 (Figure 1, time point A). However, an unprecedented surge of infections hit Taiwan again in April 2022, the culprit this time being the Omicron variants (Figure 1, time point B). The three most common lineages of Omicron are identified as BA.1, BA.2 and BA.3. Although Omicron generally causes less severe disease than its predecessor, a large volume of cases in a community in a short time period could potentially overwhelm the healthcare system.

Under the airway consensus and guidelines, video-laryngoscopy is recommended as the first-choice tracheal intubation tool during the COVID-19 pandemic [1,2,3]. Meanwhile, personal protective equipment (PPE) is essential and required for airway managers (operators) who care for the COVID-19 patients [4,5]. Initially a variety of improvised devices were designed to protect the airway managers from contaminations, such as the aerosol box, which was designed as a barrier enclosure for tracheal intubators [6]. Unfortunately, it was quickly shown that such boxes were potentially more harmful to both patients and the medical staff [7,8]. We proposed applying a piece of clear plastic sheet as a drape along with a video-assisted intubating stylet for tracheal intubation to protect the intubator from exposure to the aerosol and/or secretions during intubation (Figure 2 and Figure 3) [9,10,11,12]. The clear and soft plastic sheet adds an additional layer of protection against the possible splash of secretion/mucus, without interference or hindering of the intubation procedure [13]. In this report, we share our own clinical experience of applying the Shikani video-assisted intubating stylet technique [14,15,16,17] with a plastic sheet that acts as an ancillary barrier in patients during the Omicron pandemic in Taiwan (Figure 1, time point B).

In the following case presentations, the tracheal intubation was performed by a senior anesthesiologist with many years of experience in laryngoscopy and the Shikani video-assisted intubating stylet technique. The assistants were the residents from the Department of Anesthesia. It should be mentioned that, in this 1110-bed hospital with 20,000 anesthesia cases annually, more than 90% of tracheal intubations were conducted using the above technique (i.e., more than 7000 cases a year).

## 2. Representative Cases

From 1 April to 30 April 2022, due to hospital-wide Omicron infection, all in-hospital medical routine and elective surgical operations were suspended at our institution, with only emergency and urgent surgical procedures being performed. The surgery volume decreased by 90%. The tracheal intubators were recommended to use optimal barriers against any saliva/mucus spread or spill from patients’ mouths and airways during induction of anesthesia in addition to wearing PPE. All of the following 20 cases in this report were performed by the senior anesthesiologist using the video-assisted intubating stylet technique and plastic drape (Figure 2 and Figure 3). A technical note describing how to fabricate and prepare a plastic sheet as a barrier has been previously written [11]. Briefly, a transparent and soft plastic sheet (e.g., excised from a plastic trash bag made of ethylene vinyl acetate, 0.05–0.10 mm in thickness; 50 × 80 cm^2^ in size) was prepared. Two separate holes were made to accommodate the intubating stylet and suction, respectively, on the drape shown in Figure 2 and Figure 3 (58 × 58 cm in size, Yao I Fabric Co., Ltd., Hemei Township, Changhua County, Taiwan). The patient was preoxygenated with the mask and the circuit connected to the anesthesia machine. After the patient was anesthetized and ready for tracheal intubation, the mask was removed and then the stylet-endotracheal tube was introduced through the premade perforation of the drape. After the endotracheal intubation was confirmed, the plastic sheet was carefully split along a seamline, inverted, and removed. Caution was made to ensure no further contamination from the plastic drape. The endotracheal tube was secured according to institution standard.

The first patient (case A in Table 1) was a 44-year-old woman (159 cm/81 kg, BMI 32.0 kg/m^2^) scheduled to undergo DaVinci robotic-assisted laparoscopic myomectomy and ovarian cystectomy (Figure 4). General anesthesia was induced with routine medications, including glycopyrrolate, midazolam, fentanyl, lidocaine, propofol, and rocuronium. A 7.0 mm endotracheal tube was introduced into the upper airway using the Shikani video-assisted intubating stylet technique. The intubation process was smooth and shown in Figure 4 and in a video clip in the Appendix A.

The second patient (case B in Table 1) was a 35-year-old man (175 cm/76 kg, BMI 24.8 kg/m^2^) for emergency laparoscopic appendectomy (Figure 5). Anesthesia was induced with routine medications (including glycopyrrolate, midazolam, fentanyl, lidocaine, propofol, and rocuronium). Copious foamy secretions were present in the oropharynx. The epiglottis was floppy and the vocal cord was posteriorly displaced. The intubation was successful on the first attempt using the Shikani technique with a 7.0 mm endotracheal tube (video clip in the Appendix A).

Figure 6, Figure 7 and Figure 8 shows the intubation in a patient (case C in Table 1) with anticipated difficult airway. A 51-year-old man (166 cm, 66 kg, BMI 23.9 kg/m^2^) with lung abscess and pulmonary empyema was scheduled to receive video-assisted thoracic surgery (VATS). The patient had suffered from tongue squamous cell carcinoma (pT4aN2c, LVE/PNI positive) and was to receive combined surgical treatment with concurrent chemo-radiation therapy. The pre-operative airway evaluation predicted difficult airway management due to limited mouth-opening (<3 cm), upper lip overbite (Mallampati grade 3), and stiff neck secondary to radiation therapy. Copious and thick mucus secretions were seen during airway examination (Figure 6). An 8.0 mm endotracheal tube was used in anticipating difficulty intubation and placement of a double-lumen endobronchial tube. The tracheal intubation was performed using the Shikani technique (Figure 7). The intubation process was smooth and shown in Figure 8 and in the video clip in the Appendix A.

Figure 9 and Figure 10 demonstrate the Shikani technique intubation with double-lumen tube. A 51-year-old man (case D in Table 1; 169 cm, 70 kg, BMI 24.5 kg/m^2^) with pulmonary empyema (right-side lung) was scheduled for VATS. He also suffered lung adenocarcinoma (cT4N2M1c stage IVB) with brain and bone metastases. The transparent plastic sheet barrier was applied as shown in Figure 9. The left-sided double-lumen endobronchial tube (35 Fr) was placed on the 1st attempt. The intubation process was smooth and shown in Figure 10 and video clip in the Appendix A.

The last case to be presented is a 39-year-old man (case J in Table 1; 176 cm/76 kg, BMI 24.5 kg/m^2^) who had deep cutting injuries over extremities and massive bleeding. Emergency plastic surgery was scheduled under general anesthesia. A piece of plastic sheet barrier was put on before starting tracheal intubation using the Shikani technique. The intubation procedure was smooth, swift, and accurate (Figure 11 and the video clip in the Appendix A). The whole surgical procedure was smooth and was completed in 3 h.

## 3. Discussion

Healthcare utilization and services were dramatically constrained during the COVID-19 pandemic [18,19,20,21], and it became imperative to protect healthcare workers against infection when caring for COVID-infected patients. In-hospital spread of Omicron infection occurred at our institution in April 2022, after a huge surge of cases in the general population of Taiwan (Figure 1). Several measures were taken to address the pandemic, including massive antigen rapid testing and PCR screening, surveillance, containment, and reduction in non-emergency out-patient department visits and surgical operations. There was an urgent need for a safe and reliable airway management method during urgent and emergent intubations in the operating theatre, emergency rooms, intensive care units and general wards.

After the outbreak and later pandemic of COVID-19 in December 2019, several airway management consensus guidelines were proposed [1,2,3]. Adequate personal protective equipment (PPE) became a requirement for all frontline healthcare providers, including airway intubators who cared for a patient with suspected or confirmed COVID-19 infection. These include an N95 mask respirator, eye protection goggles, a face shield, hand hygiene, gloves, scrubs, protective clothing, a disposable cap, disposable shoe covers and when accessible, a powered air-purifying respirator (PAPR) [4,5]. The problem was that full gear PPE was not always available, nor affordable. Supplemental protective measures were hence needed to provide additional protection for airway management providers.

Many methods and techniques were previously described to safeguard airway intubators, including an aerosol box-like physical barrier which was supposed to decrease the amount of potential aerosol and droplets exposure [6]. Unfortunately, it was quickly found that such devices had significant limitations and even caused potential harm both to the patients and the intubators [7,8]. In this paper, we describe a simple method using the Shikani technique and an inexpensive and readily available clear plastic drape which covers the patient’s head and upper body and contains any aerosol spread during airway intubation [9,10,11,12]. Devices incorporating the use of a plastic drape and/or barrier were previously reported for provider protection during airway management and aerosol-generating medical procedures (AGMPs) [13]. Using the Shikani video-assisted intubating stylet through the drape allows the operator to continuously visualize the upper airway and glottis from a distance during the intubation without the operator having to place his/her face close to the mouth of the patient. We recommend that the provider also uses PPE, to provide further protection.

Since they were introduced decades ago, video-laryngoscopes have brought a paradigm shift for airway management, both in normal and difficult conditions [22,23]. Meta-analysis shows that video-laryngoscopy is superior to conventional direct laryngoscopy [24]. In our 1110-bed tertiary medical center in Taiwan, more than 92% of intubations are performed with the Shikani technique using a video-assisted intubating stylet. The rest of the intubations are conducted either with video-laryngoscopes or flexible fiber-optic bronchoscopes. Accordingly, we have a uniquely large clinical experience with this technique and we emphasize its advantages over video-laryngoscopy [10,12].

In this article, we present 20 clinical cases in which the video-assisted intubating technique and plastic sheet barrier were used for airway intubation during the April 2022 Omicron surge in Taiwan (the time point B in Table 1). In our opinion, this method of intubation is safe and effective, and we recommend it for managing COVID-positive patients or patients suspected to be infected with the virus, in order to avoid infection of the intubating operator. We realize that we do not have full data on the infection rates of the operator, as we did not routinely test our providers after procedures. However, our operator did not exhibit any clinical signs or symptoms of COVID infection. We also note that all intubations were carried out exclusively by one senior anesthesiologist, who was quite experienced in using the Shikani video-assisted intubating stylet. We believe that an experienced operator is optimal when intubating COVID cases, in order to expedite the procedure and avoid any trials and errors that are typically seen during a novice’s learning curve. The learning curve for the novice operators to use intubating stylet technique could vary individually and widely, from the very best (*n* = 2) to very hesitant (*n* = 10). Hence, it is recommended that the junior practitioners practice first their airway management skills in a medical simulation center (cadaver-based learning) and a clinical skill training center (mannequin-based learning), and then move to routine anesthesia cases prior to handling patients who may be infected with COVID. Even though the plastic sheet barrier is a good and sensible enclosure to prevent further contaminations from intubation (such as splashed saliva or spilled mucus from patients), its true safety and effectiveness is difficult to validate. This same problem has also been brought up for the box-barrier enclosure design [25,26,27,28]. However, Figure 12 demonstrates only qualitatively the prevention of spreading of nebulized watery smoke by the plastic sheet in a mannequin model. The size of the COVID-19 virus is around 0.1 μm in diameter, while the size of the aerosols ranges widely, from sub-microns to hundreds of microns. To date, quantitative research into the aerosol flow dynamics with or without barrier is not available. Therefore, tracheal intubators still are required to wear adequate PPE for airway management for the at-risk AGMPs. While we are unable to quantify the exact amount of aerosols that might have escaped from under the drape, we can only emphasize the importance of the fact that the “seeing” stylet helps the operator to keep his/her face at a far distance from the patient’s mouth and droplets, hence greatly enhancing safety.

In conclusion, we demonstrate that the Shikani video-assisted intubating stylet technique with a plastic sheet barrier (together with proper PPE) is a safe and effective method for tracheal intubation during the Omicron and other COVID-19 variants pandemic. We also recommend this method of intubation for all AGMPs where the operator might be exposed to potentially infectious airborne germs, including influenza, tuberculosis, methicillin-resistant *Staphylococcus aureus* (MRSA), HIV, hepatitis, Ebola and others.

## Figures and Tables

**Figure 1 healthcare-10-01105-f001:**
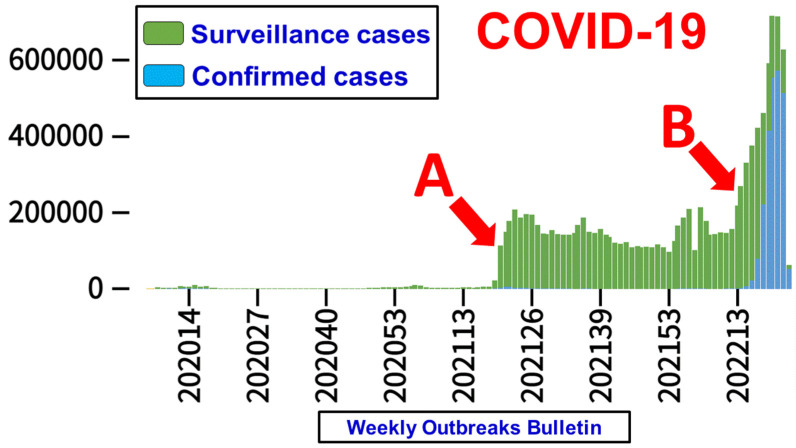
Surveillance data of COVID-19 in Taiwan. Two un-expected hits of this disease in Taiwan were noticed. (Time point **A**) The first hit occurred in May 2021 and was caused by the Alpha variant. There were 37,710 confirmed cases and 854 deaths; (Time point **B**) the second hit started in April 2022 caused by the Omicron variant (Data modified from the press release by the Taiwan Centers for Disease Control, https://www.cdc.gov.tw/En; accessed on 6 June 2022).

**Figure 2 healthcare-10-01105-f002:**
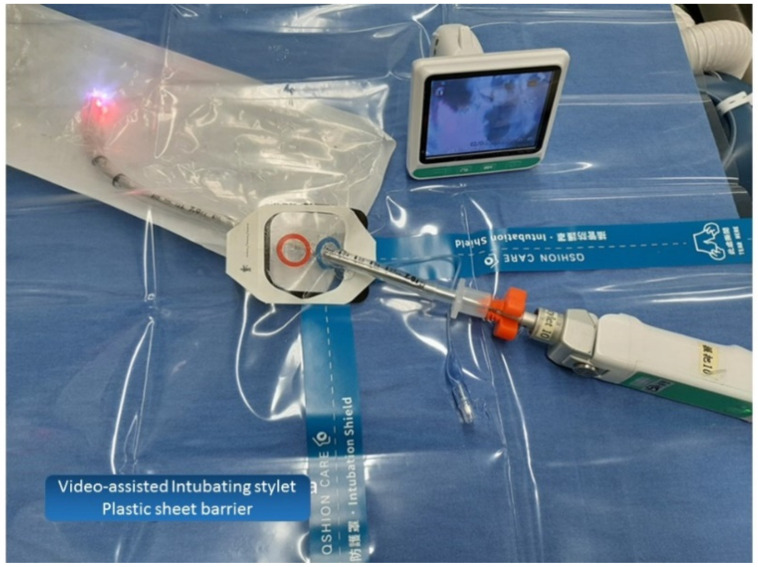
Preparation of the video-assisted intubating stylet, endotracheal tube, and a piece of manufactured plastic sheet. The technical note for preparation and specifications of the clear plastic sheet barrier can be found in the reference [11].

**Figure 3 healthcare-10-01105-f003:**
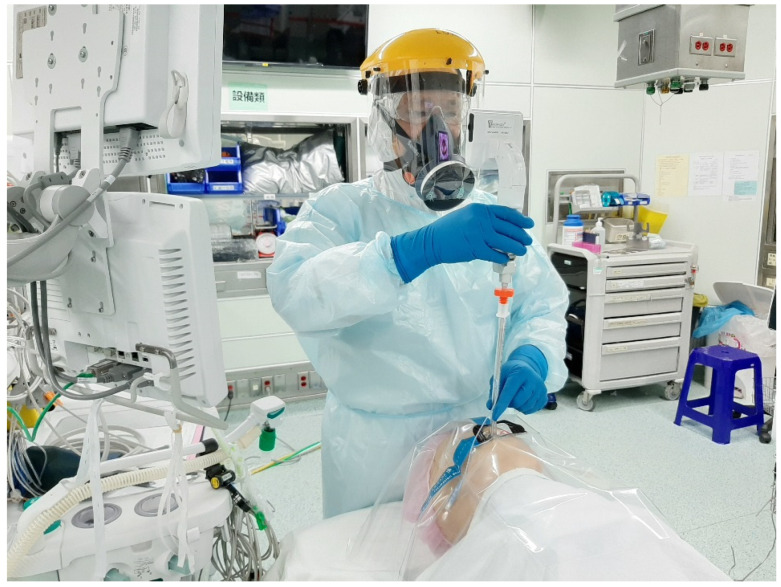
The simulation of applying the video-assisted intubating stylet technique in a mannequin covered with a clear plastic sheet over the head area and upper torso. The tracheal intubator was fully equipped with the PPE and kept a reasonably long distance away from the at-risk area of the mannequin. The technical note for preparation and specifications of the clear plastic sheet barrier can be found in the reference [11].

**Figure 4 healthcare-10-01105-f004:**
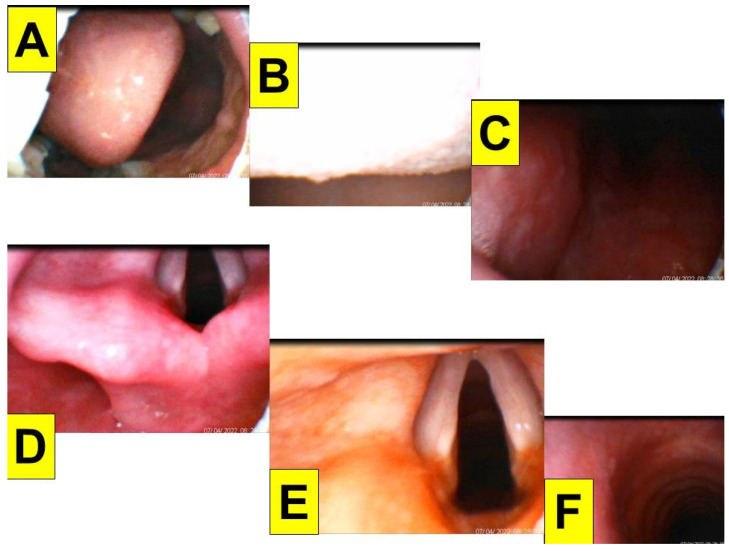
Various views displayed at the video monitor of the intubating stylet through a piece of plastic sheet barrier. (**A**) Entry of oral cavity; (**B**) partial view of tongue and oral space; (**C**) pharyngeal space; (**D**) arytenoid cartilages; (**E**) glottic view with open vocal cords; (**F**) intra-tracheal space. The time for intubation (from lip to trachea) is 8 s (see also the video clip in the Appendix A).

**Figure 5 healthcare-10-01105-f005:**
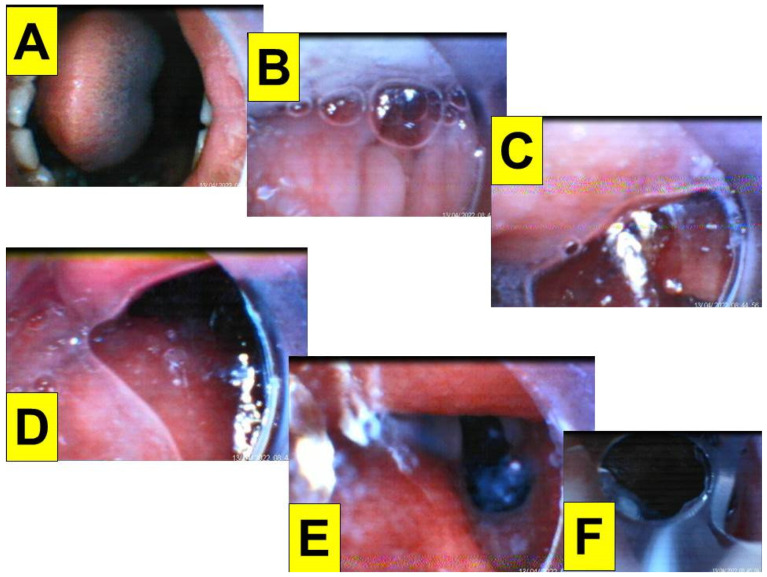
Various views displayed at the video monitor of the intubating stylet through a piece of plastic sheet barrier. (**A**) Entry of oral cavity; (**B**) partial view of tongue and oral space; (**C**,**D**) pharyngeal space and epiglottis; (**E**) glottic view with open vocal cords; (**F**) intra-tracheal space. The time for intubation (from lip to trachea) is 10 s. Copious saliva and bubbles, crowded pharyngeal space, and posteriorly displaced and relaxed epiglottis were noticed (see also the video clip in the Appendix A).

**Figure 6 healthcare-10-01105-f006:**
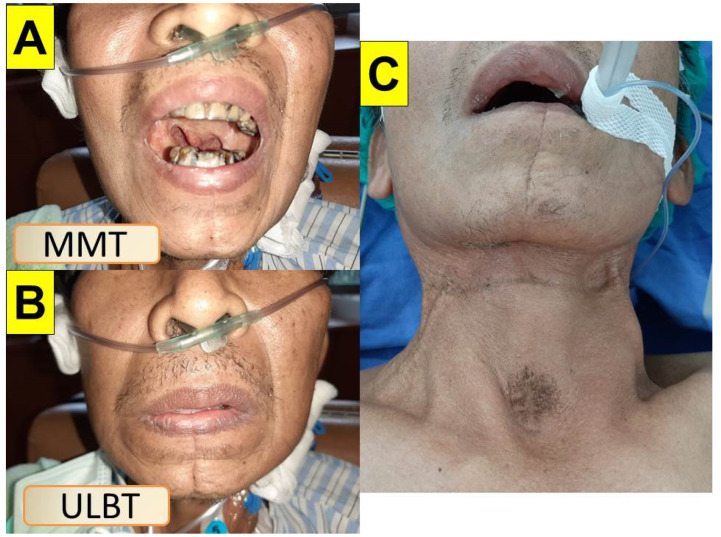
A patient was scheduled to receive video-assisted thoracic surgery (VATS) due to pulmonary empyema. He had prior history of tongue cancer and had received surgical resection, radiation therapy and chemotherapy. (**A**) Mouth-opening was about 3 cm with a grade 3 modified Mallampati test; (**B**) upper lip bite test, grade 3; (**C**) the head and neck displayed surgical scars, radiation fibrosis, and tracheostomy scar. MMT: Modified Mallampati Test. ULBT: Upper lip bite test.

**Figure 7 healthcare-10-01105-f007:**
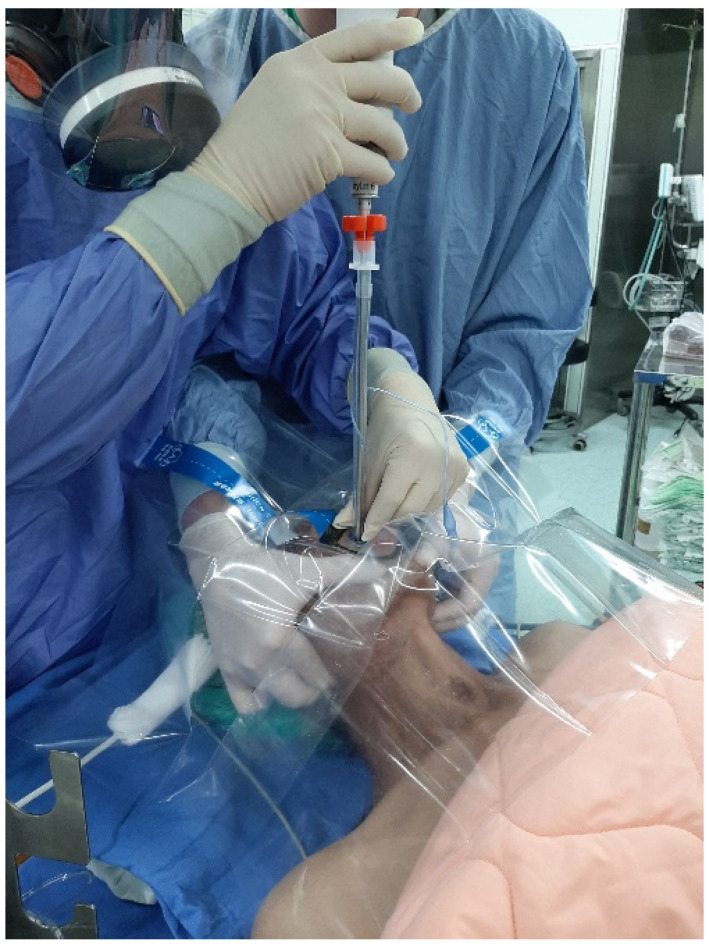
Tracheal intubation through a piece of plastic sheet barrier (The same patient as the one in Figure 6 and Figure 8).

**Figure 8 healthcare-10-01105-f008:**
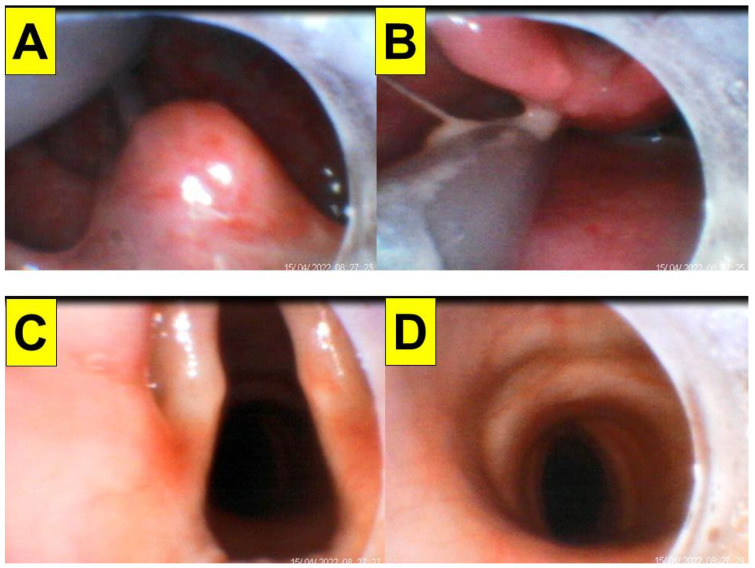
Various views displayed at the video monitor of the intubating stylet through a piece of plastic sheet barrier. (**A**) Entry into oral cavity with identified uvula; (**B**) partial view of pharyngeal-laryngeal space and arytenoid cartilages; (**C**) vocal cords; (**D**) view inside the trachea. The tip of a suction tube was seen in (**A**,**B**). The time for intubation (from lip to trachea) is 7 s (see also the video clip in the Appendix A).

**Figure 9 healthcare-10-01105-f009:**
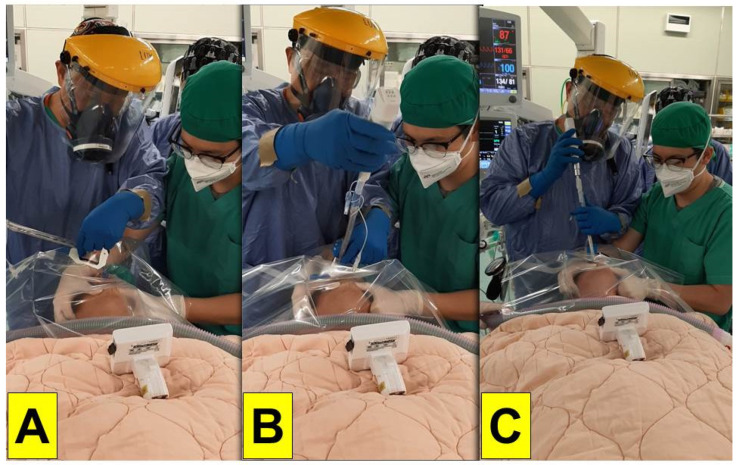
Application of the Shikani technique and plastic sheet barrier in the patient receiving double-lumen endobronchial tube insertion. (**A**) Entry of oral cavity; (**B**) localization of glottic area; (**C**) ready for railroading the tube into trachea. The pictorial and video demonstration can be found in Figure 10.

**Figure 10 healthcare-10-01105-f010:**
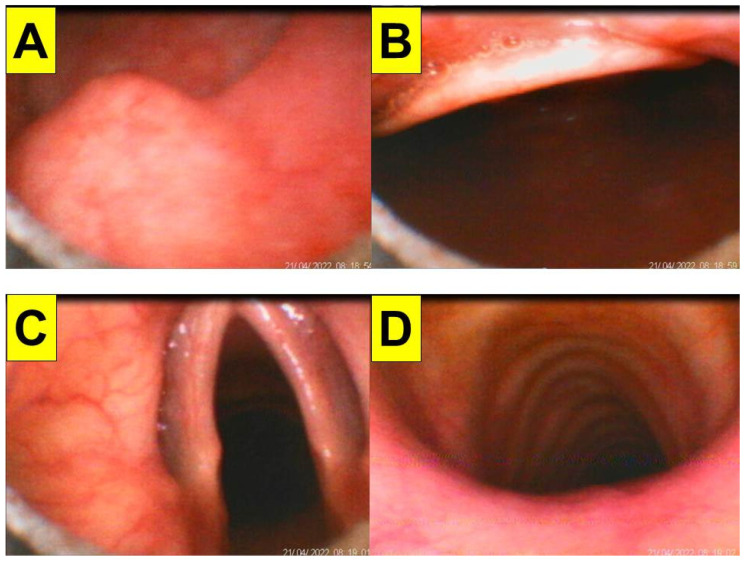
Application of the Shikani technique and plastic sheet barrier in the patient receiving double-lumen endobronchial tube insertion. (**A**) Entry into oral cavity and both the uvula and palates were seen; (**B**) localization of epiglottis; (**C**) glottic view with open vocal cords; (**D**) ready for railroading the tube into trachea. The time for intubation (from lip to trachea) is 14 s (see also the video clip in the Appendix A).

**Figure 11 healthcare-10-01105-f011:**
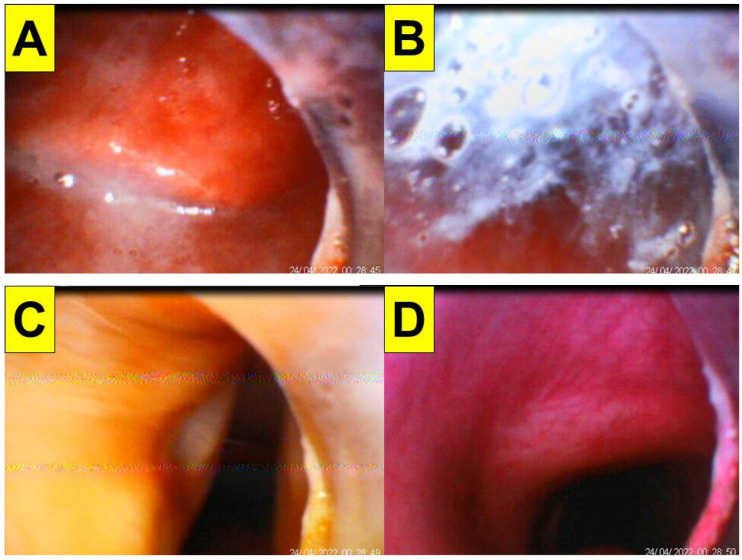
Tracheal intubation using the Shikani technique and plastic sheet barrier in a COVID-19 confirmed case for an emergency surgery. (**A**) Posteriorly displaced epiglottis; (**B**) obscured view by copious mucus; (**C**) full glottic view; (**D**) smooth insertion into tracheal lumen. The time for intubation (from lip to trachea) is 12 s (see also the video clip in the Appendix A).

**Figure 12 healthcare-10-01105-f012:**
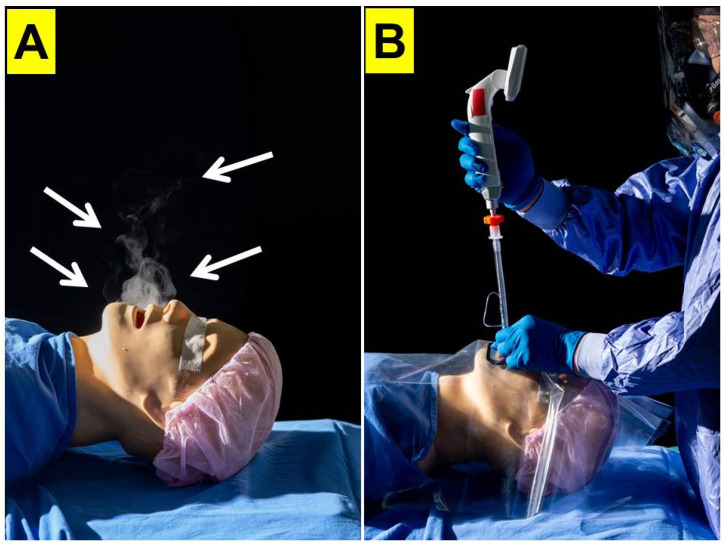
Protection from aerosol spreading contamination by video-assisted intubating stylet technique and a piece of plastic sheet barrier demonstrated in a mannequin. (**A**) Arrows in white color demarcate the area of simulated aerosol spreading; (**B**) with the coverage of a piece of plastic sheet as barrier during stylet intubation procedure, there was no aerosols detected which continuously spread under the plastic coverage.

**Table 1 healthcare-10-01105-t001:** Application of the Shikani technique and plastic sheet barrier for tracheal intubation during the Omicron pandemic (April 2022).

	Case A	Case B	Case C	Case D	Case E	Case F	Case G	Case H	Case I	Case J
Rapid antigen test/PCR test	Negative	Negative	Negative	Negative	Negative	Negative	Negative	Negative	Positive	Positive
Surgery type	Elective	Emergent	Elective	Elective	Emergent	Elective	Elective	Elective	Emergent	Emergent
Age/gender	44/F	35/M	51/M	51/M	28/F	74/M	61/F	37/M	68/M	39/M
Height (cm)/weight (kg); (BMI, kg/m^2^)	159/81; 32.0	175/76; 24.8	166/66; 23.9	169/70; 24.5	160/59; 23.0	155/60 24.9	155/59; 24.5	170/65 22.4	157/69; 27.9	176/76; 24.5
First attempt success	Yes	Yes	Yes	Yes	Yes	Yes	Yes	Yes	Yes	Yes
Number of attempt	1	1	1	1	1	1	1	1	1	1
Time to intubate	**8 s**	**10 s**	**7 s**	**14 s**	**6 s**	**6 s**	**7 s**	**9 s**	**43 s**	**12 s**
Laryngeal and glottic view	Full	Full	Full	Full	Full	Full	Full	Full	Full	Full
Degree of difficulty for intubation	Easy	Easy	Easy	Easy	Easy	Easy	Easy	Easy	In a spin ^#^	Easy *
Need to Require additional external maneuvers	No	No	No	No	No	No	No	No	No	No
Complications (soft tissues or dental injuries)	Nil	Nil	Nil	Nil	Nil	Nil	Nil	Nil	Nil	Nil
Intubating device	T	T	T	T	T	T	T	T	T	T
	**Case K**	**Case L**	**Case M**	**Case N**	**Case O**	**Case P**	**Case Q**	**Case R**	**Case S**	**Case T**
Rapid antigen test/PCR test	Negative	Negative	Negative	Negative	Negative	Negative	Negative	Negative	Negative	Negative
Surgery type	Emergent	Emergent	Elective	Emergent	Elective	Elective	Elective	Elective	Elective	Elective
Age/gender	65/M	53/F	65/F	30/F	75/F	88/F	67/M	57/F	70/M	52/M
Height (cm)/weight (kg); (BMI, kg/m^2^)	164/99; 36.8	158/60; 24.0	153/59; 25.2	170/68; 23.5	160/80; 31.1	150/56; 24.8	169/76; 26.6	160/59; 23.0	158/63; 25.2	168/82; 29.0
First attempt success	Yes	Yes	Yes	Yes	Yes	Yes	Yes	Yes	Yes	Yes
Time to intubate (from lip to trachea)	**11 s**	**6 s**	**7 s**	**8 s**	**8 s**	**7 s**	**42 s**	**42 s**	**58 s**	**6 s**
Laryngeal & glottic view	Full	Full	Full	Full	Full	Full	Full	Full	Full	Full
Degree of difficulty for intubation	Easy	Easy	Easy	Easy	Easy	Easy	In a spin	In a spin	In a spin	Easy
Need to Require additional external maneuvers	No	No	No	No	No	No	No	No	No	No
Complications (soft tissues or dental injuries)	Nil	Nil	Nil	Nil	Nil	Nil	Nil	Nil	Nil	Nil
Intubating device	T	T	T	T	T	T	S	S	S	T

BMI: Body mass index (kg/m^2^). ^#^ Operated in the negative pressurized isolation room; difficulty jaw-thrusting due to oro-facial lesions; pharyngeal view obscured by blood, mucus and collapsed soft tissues; the first-year resident serving as the airway assistant. * Operated in the negative pressurized isolation room; poor jaw-thrusting initially; the third-year resident serving as the airway assistant. T: Video Intubation Light Stylet (Trachway, Markstein Sichtec Medical Corp., Taichung, Taiwan). S: C-MAC VS (Video Stylet) (KARL STORZ SE & Co. KG, Tuttlingen, Germany). The glottic view quality was judged by either the Cormack–Lehane grades or POGO scores. The degree of difficulty for intubation was evaluated subjectively and confirmed by reviewing the video-recording images. Post-operative sore throat (POST) and hoarseness were not evaluated in all the cases.

## Data Availability

Not applicable.

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
