# Peer review of "Video-Assisted Stylet Intubation with a Plastic Sheet Barrier, a Safe and Simple Technique for Tracheal Intubation of COVID-19 Patients"

_healthcare, 2022, doi:10.3390/healthcare10061105_

Round 1

Reviewer 1 Report

Congratulation to Authors for putting together a very interesting report.

Please find below my comments regarding the manuscript:

Introduction:

Given a format of brief report introduction section could be more focus on the airway management in COVID patients.

There are some controversies regarding classification of intubation as AGP is https://doi.org/10.1111/anae.15292. It is rather circumstances patients acutely unwell, high viral load, coughs. Elective intubation itself with proper NMBA might not be as aerosol generating as we initially thought.

For the readers not familiar with this intubation stylet might be beneficial to briefly mention properties of this device.

Table 1.

I suggest that Type of surgery would be elective or emergency

First attempt success. Number of attempts, additional maneuvers, complications and intubation device could be just described in the manuscript as they were no different between cases. This will allow for better flow of the table. I would also suggest changing “clumsy” to difficult.

 Discussion

Essential point brough by authors about safety of intubation barriers initially used in the pandemic.

Couldn’t agree more.

Overall, it is very interesting and well-balanced discussion.

What is authors opinion on use of plastic sheets during RSI with a high risk of aspiration?

What additional benefits can a plastic sheet bring given than there is no shortage of PPE and the operator can wear appropriate PPE?

This a very unique experience of center that predominantly use intubation style and well described  additionally supported with figures and supplementary videos -  in mine opinion would be of much interest to readers.

Author Response

Ms Title: Video-assisted intubating stylet with a plastic sheet….

Ms Number: Healthcare, 1748415

Ms authors: Huang et al.

20220607

Responses to the Reviewer-1: Thanks for your excellent comments and professional opinions. We really appreciate your efforts and devotion.

Point-1: Introduction: Given a format of brief report, introduction section could be more focus on the airway management in COVID patients.

Response: Yes, we agree with you. Therefore, we cited the context of the guidelines of airway management during COVID-19 pandemic (references 1-3) for background introduction. Thank you.

Point-2: There are some controversies regarding classification of intubation as AGP is https://doi.org/10.1111/anae.15292. It is rather circumstances patients acutely unwell, high viral load, coughs. Elective intubation itself with proper NMBA might not be as aerosol generating as we initially thought.

Response: Thanks for your referring the article (Brown et al., Anaesthesia. 2021 Feb;76(2):174-181.) We agree with you about the impact of the intubating condition on COVID-positive patients, e.g., under sedation and profound neuromuscular block, on possible contamination to the airway operators. In particular, extubation even put higher risks to the airway managers. The plastic sheet barrier, in this case, has been suggested to be helpful not only at intubation phase but also at extubation. We would like to emphasize, however, the medical providers should always wear PPE during AGMPs to prevent possible contaminations (either from the patient or the environment).

Point-3: For the readers not familiar with this intubation stylet might be beneficial to briefly mention properties of this device.

Response: We appreciate your suggestion. We have presented figures 2 and 3 to illustrate the intubating stylet technique and also cited the related references (ref 9-12, 16,17). Actually, we have been preparing a review article about the role of intubating stylet technique and will submit it shortly. It will cover a rather broad background introduction and detailed description of intubating stylet technique in that review article.  

Point-4: Table 1. I suggest that Type of surgery would be elective or emergency. First attempt success. Number of attempts, additional maneuvers, complications and intubation device could be just described in the manuscript as they were no different between cases. This will allow for better flow of the table. I would also suggest changing “clumsy” to difficult.

Response: Thank you for the excellent suggestion. We have therefore revised the Table 1 accordingly. “Surgery type” has been revised and only indicated either “emergent” or “elective”. Also, “clumsy” has been changed to “in a spin”. Although “easy and difficult” is the normal expression to describe the intubating process, here we would like to emphasize, in these several cases, it was not difficult to intubation by definition. Instead, it was kind of awkward during intubation related to using C-MAC VS itself.

Point-5 : Discussion section: Essential point brought by the authors about safety of intubation barriers initially used in the pandemic. Couldn’t agree more.

Response: Various improvised intubation barriers (e.g., aerosol box) have been designed and innovated during the early phase of COVID-19 pandemic. However, several safety issues about such enclosure barrier have been brought for discussion (e.g., “More on Barrier Enclosure during Endotracheal Intubation” published in NEJM, DOI: 10.1056/NEJMc2012960). Thanks for your comments. 

Point-6: What is authors’ opinion on use of plastic sheets during RSI with a high risk of aspiration?

Response: This is a more complicated scenario indeed. When COVID-19 patient need tracheal intubation and meanwhile with high risk of aspiration (e.g., full-stomach), the airway management with RSI under such condition definitely is also very challenging. The benefit of using plastic sheet then is even more outstanding because it could prevent contamination from the patient’s vomitus or puke. Actually, therefore are few such contaminated cases occurred in Taiwan during the pandemic. The airway management team members contracted COVID from the patient’s vomitus while tracheal intubation was proceeded. More important, the use of plastic sheet would not interfere or hinder the RSI per se.

Point-7: What additional benefits can a plastic sheet bring given that there is no shortage of PPE and the operator can wear appropriate PPE?

Response: Excellent point! During the pandemic of COVID-19, the healthcare resources inequity has been brought to attention for the general public in the global village. However, if the shortage of PPE is not a prominent issue in certain countries/regions and the airway operators worn appropriate PPE during AGMPs, then in our opinion, it will be fine and safe for them to perform the task. Meanwhile, even under such condition, the ancillary role of plastic sheet is to provide another layer of safety to prevent the possible splash or spill-over of the secretion, mucus, saliva and possible blood from patient’s mouth (e.g., when patient is coughing or bucking).

Point-8: This is a unique experience of center that predominantly use intubation style and well described-additionally supported with figures and supplementary videos. In my opinion this would be of much interest to readers.

Response: Thank you very much. We also are eager to share our clinical experiences of using video-assisted intubating stylet technique with the professional medical communities.

Reviewer 2 Report

Thank you for the opportunity to review this interesting case series on video-assisted stylet-intubation under COVID19 conditions. The manuscript ist presented in a clear and comprehensible manner, figures and tables appear in high quality, and the reference citations are appropriate. I have only few minor suggestion/comment for the authors:

#How are higher volumes of saliva or reflux of gastric content managed during laryngoskopy? I would find it interesting how suction catheters are being placed in these special events. 

#Blurring and fogging up may be a limitation in this video-assisted approach. How do the authors carry out preventive measures or specific maneuvers to clean up the device?

#Although the authors already mentioned that the success rates depend on experience and training, readers might found it interesting how the average learning-curve is for novice users.

I would be happy to read the response of the authors in a short revision of this manuscript. Best wishes and good luck!

Author Response

Responses to the Reviewer-2: Thanks for your excellent comments and professional opinions. We really appreciate your efforts and devotion.

Thank you for the opportunity to review this interesting case series on video-assisted stylet-intubation under COVID19 conditions. The manuscript is presented in a clear and comprehensible manner, figures and tables appear in high quality, and the reference citations are appropriate. I have only few minor suggestion/comment for the authors:

Response: Thank you for your expert opinion and constructive comments on our manuscript. We therefore responded to your comments as follows:

Point-1: How are higher volumes of saliva or reflux of gastric content managed during laryngoscopy? I would find it interesting how suction catheters are being placed in these special events.

Response: This is a crucial issue and appreciate your bringing it up for discussion. One of the troubles intubating style technique faced is the interference by the saliva, secretion, and even blood and vomitus. There is no way out except adequate suction. One should always prepare Yankauer suction tip and other suction tubes at hand during airway management. In particular, we always use suction tube to clear the airway and left there before starting inserting the intubating stylet. This simple straight maneuver is really helpful for video-assisted intubating stylet technique.

Point-2: Blurring and fogging up may be a limitation in this video-assisted approach. How do the authors carry out preventive measures or specific maneuvers to clean up the device?

Response: This is a good and practical question and very often the fogging issue really disturbed the airway operator if optical intubating tool is used for tracheal intubation. Several traditional methods to handle this problem include using defogging solution, tipping the stylet in the warming solution, etc. To clean up the dirty lens is the only way and important way if it happened.

Point-3: Although the authors already mentioned that the success rates depend on experience and training, readers might found it interesting how the average learning-curve is for novice users.

Response: This is an interesting question and needed to be answered. Actually, we had studied years ago about the learning curve of using intubating stylet (compared with video-laryngoscopy and direct laryngoscopy), using the mannequin model. Also we have observed the performance of novice operators (PGY students) and found the learning curve is from n=2 to n=10. We would like to emphasize, however, various human factors determined the learning curve of using intubating tools.

Point: I would be happy to read the response of the authors in a short revision of this manuscript. Best wishes and good luck!

Response: We really appreciate your constructive comments on our works.
